

# Comparison of carbon content between plantation and natural regeneration seedlings in Durango, Mexico

Jesús Alejandro Soto-Cervantes[1], José Javier Corral-Rivas[2], Pedro Antonio Domínguez-Calleros[2], Pablito Marcelo López-Serrano[2], Eusebio Montiel-Antuna[2], Emily García-Montiel[2] and Alberto Pérez-Luna[3,4]

[1] Programa Institucional de Doctorado en Ciencias Agropecuarias y Forestales, Universidad Juárez del Estado de Durango, Durango, Durango, México
[2] Facultad de Ciencias Forestales y Ambientales, Universidad Juárez del Estado de Durango, Durango, Durango, México
[3] Centro de Bachillerato Tecnológico Industrial y de Servicios Número 89, Dirección General de Educación Tecnológica Industrial, Durango, Durango, México
[4] Postgrado en Ciencias Forestales, Colegio de Postgraduados, Texcoco, México, México

Corresponding author
Alberto Pérez-Luna, aperez@ujed.mx

## ABSTRACT

Forest plantations and natural forests perform a relevant role in capturing $CO_2$ and reducing greenhouse gas concentrations. The objective of this study was to compare the diameter increment, biomass and carbon accumulation in a plantation of *Pinus durangensis* and a naturally regenerated stand. The data were collected from 32 circular plots of 100 m$^2$ (16 plots in the planted site and 16 in naturally regenerated area). At each plot, the diameter at the base (cm) and height (m) of all seedlings were measured using a Vernier and tape measure, and a seedling was destructively sampled collecting one cross-section at the base of the stump. The annual ring-width increment of each sampled seedling was recorded to obtain its diameter at the base and estimate annual aboveground biomass and carbon accumulation through allometric equations. The response variables were evaluated using mixed-effects ANOVA models. Results indicated that there were significant differences ($P \leq 0.05$) on annual tree-ring width growth, biomass and carbon accumulation. The plantation seedlings showed significantly higher growth rates, biomass and carbon accumulation at most evaluated years. After 7 years of growth the lines of current annual increment (CAI) and mean annual increment (MAI) in basal diameter for both the plantation and the natural regeneration have not yet intersected. Both forest plantations and naturally regenerated stands of the studied tree species may be suitable alternatives to promote $CO_2$ capture and increase timber production.

## INTRODUCTION

Plantations and natural forests are an excellent alternative for $CO_2$ capture and to reduce greenhouse gas concentrations and to mitigate the effects of global warming (*Patiño Forero et al., 2018*; *González-Cásares et al., 2019*).

Mexico is one of the highest emitters of greenhouse gases, which include 407,695 megatons of $CO_2$ annually (*INECC-SEMARNAT, 2015*). Therefore, it is important to know the stocks and flows of carbon in its forest areas. Biomass estimation can be calculated using indirect methods such as the use allometric equations (*Návar et al., 2004*; *Montes de Oca-Cano et al., 2008*; *Vargas-Larreta et al., 2017*), and remote sensing techniques (*López-Serrano et al., 2015*).

*Pinus durangensis* Martínez is one of the most commercially important tree species in northern Mexico. It is also one of the forest species with the greatest potential to accumulate carbon (*Graciano-Ávila et al., 2019*). Its performance and adaptability allow it to be used frequently in commercial forest plantations in the state of Durango (*Prieto Ruíz et al., 2016*). *Flores, Pineda Ojeda & Flores Ayala (2019)* estimate that in Mexico, 1,400 hectares of *Pinus durangensis* are planted annually. The success of the forest plantations depends to a large degree on the use of high-genetic-quality plants (*Stewart et al., 2016*) with good adaptability to the plantation site (*Grossnickle, 2012*; *Vallejo et al., 2012*). In Mexico, 57% of mortality in plantations is caused by poor-plant quality (*Prieto Ruíz et al., 2016*), in addition to bad practices that occur during the plantation process (*Burney et al., 2015*).

With the establishment of commercial forest plantations in Mexico, the aim is to increase the country's timber production (*Hernández-Zaragoza et al., 2019*). They also constitute an important element in carbon sequestration and the generation of other environmental services. However, there are few studies related to the evaluation of their timber yield and their potential for carbon sequestration (*Soto-Cervantes et al., 2020*).

The natural regeneration of pine species by means of the use of seed-tree cuttings or selective fellings are the most used methods to achieve the establishment of the new seedlings in the areas under forest management in the Mexican temperate forests (*Ramírez Santiago et al., 2015*). In addition, they are appropriate options for the ecological rehabilitation of forests (*Pensado-Fernández et al., 2014*), since they have the advantage to retain the germplasm's genetic memory (genotype) to guarantee a higher percentage of survival and development than plantations (*Landis, 2011*). However, few studies have compared tree growth and carbon sequestration between forest plantations and natural regeneration of *Pinus durangensis* (*Fernández-Pérez, Ramírez-Marcial & González-Espinosa, 2013*).

As a hypothesis of this study, we suppose that there are significant differences ($P < 0.05$) in terms of the increase in the diameter at the base and carbon capture between plantation and natural regeneration seedlings. Therefore, the objectives of this work were (i) to compare the basic bioclimatic conditions (precipitation and temperature) of the study areas in the years of growth of the seedlings evaluated, (ii) to compare the diameter increment at the base, (iii) estimate and compare the accumulated content of biomass and

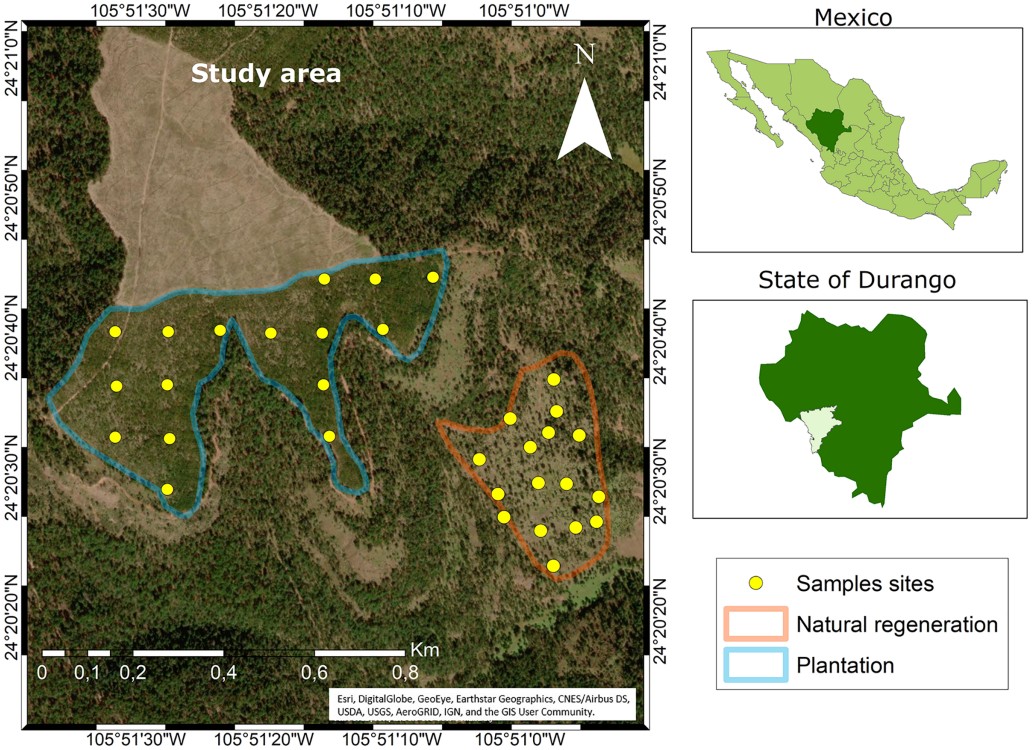

**Figure 1** **Location of the study area.** Source: ARCGIS Software (at Esri, DigitalGlobe, GeoEye, Earthstar Geographics, CNES/Airbus DS, USDA, USGS, AeroGRID, IGN, and the GIS User Community). Maps were created using ArcGIS® software by Esri. Copyright ©Esri. All rights reserved. Available at https://www.arcgis.com/index.html#.

carbon and (iv) estimate the current annual increment (CAI) and mean annual increment (MAI) in basal diameter of *Pinus durangensis* seedlings in a plantation and a stand of natural regeneration.

# MATERIALS AND METHODS

## Characteristics of the study area

The study area is located on the private property "Las Veredas" in the municipality of San Dimas, in Durango, Mexico (Fig. 1). Data collection was carried out in two sub-stands: (1) clear-cutting with immediate plantation of 21.4 ha located at coordinates 24°20′40″ N and 105°51′20″ W with an elevation of 2,600 m, and (2) seed-tree cutting of 10.2 ha located at 24°20′30″ N and 105°50′58″ W at 2,800 m altitude.

The climate in the study area is temperate with summer rains (CW) and the temperature ranges between −3 and 18 °C (*García, 2004*). The average slope in the clearcut is 9%, while in the seed-tree cutting is 27%. The most frequent rains occur between July and September, which provide an average annual rainfall of 1,000 mm. The soil of both stands is similar and may be classified as Umbrisol (*INEGI, 2007*). The characteristic vegetation is mixed coniferous and broadleaf forests. The first type is dominated by pines, of which the following stand out: *Pinus durangensis*, *P. cooperi* C.E. Blanco, *P. teocote* Schl. et Cham. and *P. strobiformis* Engelm. Of the broadleaves, the oaks stand out, among these:

**Table 1 Descriptive statistics of the study sites and sampled seedlings.**

| Study sites | Area (ha) | Stand variable | | | | Sampled seedlings | | | | | |
| | | N | | BA (m²) | | Age (years) | | DB (cm) | | Height (m) | |
| | | Mean | SD | Mean | SD | Mean | SD | Mean | SD | Mean | SD |
| Plantation | 21.40 | 1,881 | 389 | 7.41 | 2.85 | 8 | 0.00 | 8.20 | 1.06 | 3.70 | 0.50 |
| Natural Regeneration | 10.20 | 725 | 420 | 10.58 | 10.01 | 11 | 1.45 | 8.36 | 2.11 | 4.50 | 1.20 |

Note:
  N, Number of trees per hectare; BA, Basal area per hectare (m²); DB, diameter at the base; SD, Standard deviation.

*Quercus rugosa* Née and *Q. sideroxyla* Bonpl. There are also trees of some ecologically important species of the genera *Juniperus*, *Arbutus* and *Alnus*, among others (*González-Elizondo et al., 2012*).

## Forestry background

The clearcut was carried out in February 2010, and during the rainy season (July and August) of the same year the plantation of *Pinus durangensis* was established.
The seedlings were 12 months old at the time the plantation, which was carried out with a density of 2,500 plants/ha. The plantation was produced with germplasm from trees with superior genetic characteristics that grow in seed tree stands near the study area. Land preparation consisted of clearing, scattering and doing a controlled burn of forest waste. To improve soil conditions, the soil was plowed using a D-6 track-type tractor equipped with a ripper. On the other hand, the seed-tree cutting was carried out in 2007 with a cut intensity of standing trees of 58% for the genus Pinus, 100% for Quercus, 100% for Juniperus and 100% for Arbutus, leaving approximately 25 fifty-year-old mother trees per hectare of *Pinus durangensis* to promote the natural regeneration.

The monthly climatic data of total precipitation, as well as the maximum, minimum and mean temperatures for the grid centered on the study area (24–24.5°N, 105–105.5°W) were obtained through the KNMI-Climate database (*Van Oldenborgh & Burgers, 2005*) as plantation growth and natural regeneration in the early years occurred at different times.

## Field evaluation

To collect the data, 32 circular plots of 100 m² were established (16 in the plantation and 16 in the seed-tree cutting). These were distributed through a completely randomized design according to the methodology established by *CONAFOR (2013)*. Between the plantation and the natural regeneration, there is a distance of approximately 350 m. At each plot, variables such as diameter at the base (cm) and height (m) were measured using a Vernier and a 5 m tape measure, respectively. A reference tree was selected according to the plot´s mode diameter and destructively sampled from each plot. The sampling of the felled seedlings involved the collection of one cross-section at the base of the stump. In total, 32 *Pinus durangensis* seedlings were felled (16 in the plantation and 16 in the stand with seed-tree cutting). Table 1 presents the descriptive statistics of the studied sites and sampled seedlings.

The cross-sections were labeled, dried and polished, and each growth ring was subsequently measured using a stereomicroscope with micrometer precision in the dendroecology laboratory at the Facultad de Ciencias Forestales y Ambientales, Universidad Juárez del Estado de Durango. The average ring width was estimated by the average of the measurement in four directions. For the comparison of ring width, only the first 7 years of growth were taken into account since the age of the seedlings in both studied sites was different. The data was obtained in July 2018; therefore, the measurement of ring widths was done retrospectively from the year 2017 because the last ring of the plantation was still under development.

## Annual biomass estimation and carbon content

The aboveground seedlings biomass was estimated with the equation developed and recommended by *Návar et al. (2004)* for *Pinus durangensis* for trees with basal diameters ranging from 5 to 15 cm. It uses the basal diameter as the predictive variable (Eq. (1), $R^2 = 0.86$).

$$Y_i = a(DB)^b \tag{1}$$

where:
$Y_i$ = Total dry biomass
a = 0.0199 y b = 2.5488
DB = Diameter at the base of the tree (cm)

Annual values of aboveground tree were derived from the difference of total biomass values between two consecutive years. The carbon content was calculated according to the percentage indicated by *Hernández-Vera et al. (2017)*, who reported a concentration of 50.36% of the total biomass *Pinus durangensis*.

## Tree increment estimation

The increment for both studied kind of seedlings was assessed on the cross-sections collected at the base of the sampled trees. The individual tree-ring width growth, individual tree current annual diameter increment (CAI), and mean annual diameter increment (MAI) were accounted for years 1 to 7. The tree-ring width growth corresponded with the individual tree annual ring width average (mm), CAI represented the individual tree basal diameter increment observed for each of the studied years (cm/year), while the MAI was the total individual tree diameter increment up to a given age divided by that age (cm/year) (*Cardalliaguet et al., 2019*).

## Statistical analysis

Due to the use of hierarchical data obtained from the same trees, differences in tree-ring width growth, biomass and the carbon accumulation among seedlings of studied sites were evaluated with a two-way mixed-effects analysis of variance model (*Snedecor & Cochran, 1989*; *Oberfeld & Franke, 2013*). In our data, the mixed-effects models attempt to generalize results beyond both the sampled sites and seedlings. Thus, to take into account the possible temporal correlation between data from the same individual, the studied sites (*i.e.*,
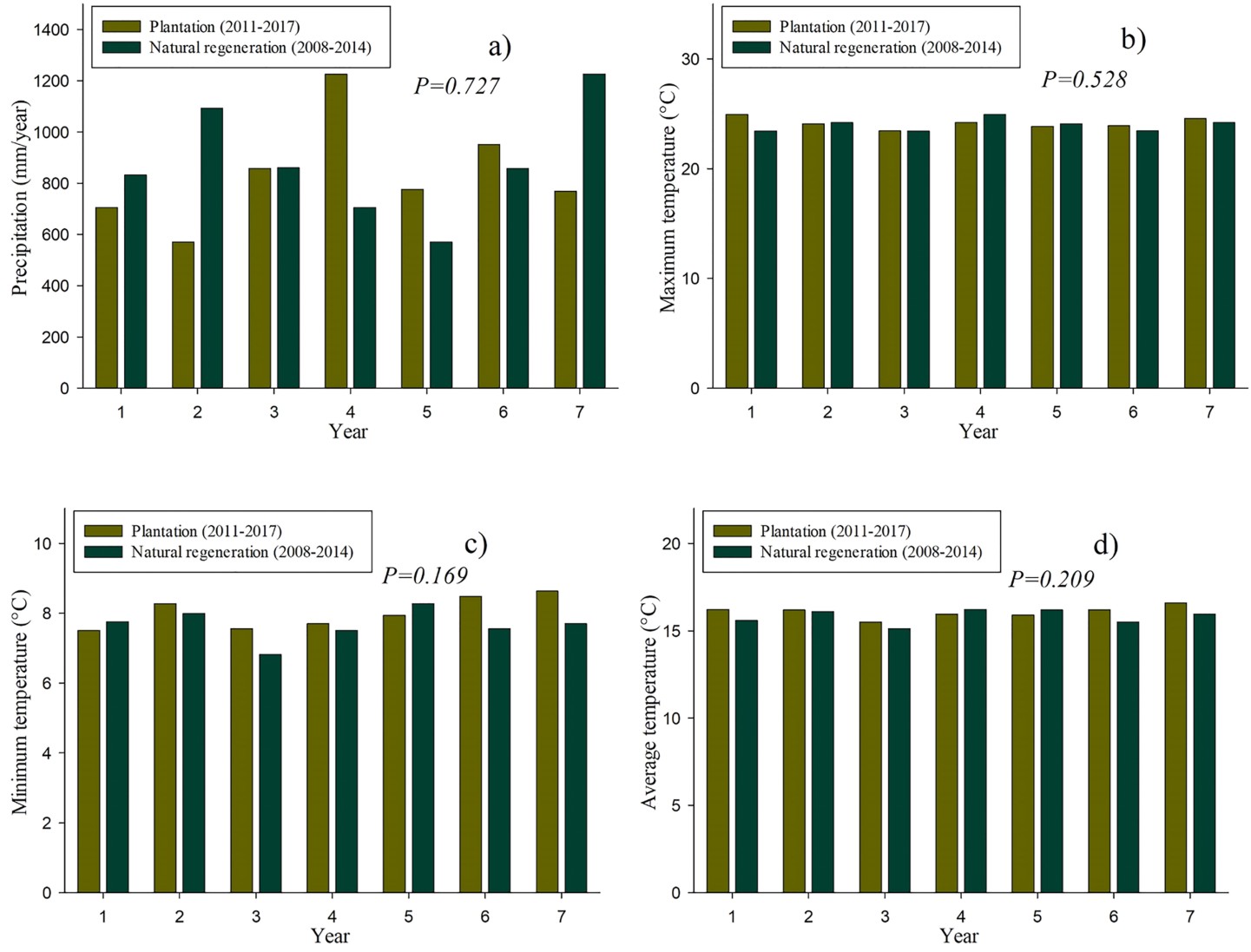

**Figure 2 Historical climate data on the plantation (2011–2017) and natural regeneration (2008–2014): Precipitation (A), maximum of monthly average temperature (B), minimum of monthly average temperature (C) and average temperature (D).**

plantation or natural regeneration) and evaluated years (1 to 7) were classified as the fixed factors, while the basal diameter of trees at age $i$ was used as a random factor. Multiple comparisons were conducted when there were significant differences among means of plantation and natural regeneration seedlings with the Tukey Means Comparison Test ($P < 0.05$). All statistical analyses were performed in the statistical software R® version 3.5.3 (*R Core Team, 2019*).

## RESULTS

Figure 2 shows the historical data of the precipitation and temperature that occurred both in the plantation and natural regeneration sites in their first 7 years. It should be noted that

**Table 2 Pairwise comparisons for annual tree-ring width growth of seedlings in plantation and natural regeneration.**

| Comparison | Year | t ratio | P value |
|---|---|---|---|
| Plantation—Natural regeneration | 1 | −0.689 | 0.4915[ns] |
| Plantation—Natural regeneration | 2 | 2.172 | 0.0312[*] |
| Plantation—Natural regeneration | 3 | 4.133 | 0.0001[***] |
| Plantation—Natural regeneration | 4 | 4.175 | <0.0001[***] |
| Plantation—Natural regeneration | 5 | 5.946 | <0.0001[***] |
| Plantation—Natural regeneration | 6 | 3.722 | 0.0003[***] |
| Plantation—Natural regeneration | 7 | 1.381 | 0.1688[ns] |

Notes:
 [*] ($P < 0.05$).
 [***] ($P < 0.001$).
 ns, not significant.

no significant differences ($P > 0.05$) were found neither in precipitation nor in temperature among the studied sites.

### Tree-ring width growth

Results of mixed-effects analysis of variance model indicated that both fixed and random factors show significant effects on annual tree-ring width growth ($P < 0.01$), which means that there are significant differences among studied sites and years when they are kept constant. Table 2 shows the pairwise comparisons for annual tree-ring width growth. There were significant differences on tree-ring width growth for most evaluated years, except at ages 1 and 7 ($P > 0.01$). Figure 3 shows that annual tree-ring width growth is significantly greater in seedlings from plantation than in trees naturally regenerated at 2–6 years. It is interesting to observe that annual tree-ring width growth of seedlings in the natural regeneration shows an upward growth tendency from the beginning to the end of the evaluation period, whereas, the annual tree-ring width growth of trees in plantation shows a decrease in growth at 6 and 7 years.

### Biomass and carbon accumulation

Results of mixed-effects analysis of variance model indicated that both fixed and random factors show significant effects on individual biomass and carbon accumulation ($P < 0.01$), which means that both biomass and carbon accumulation differ among sites and annual rates when they are kept constant. Table 3 shows the pairwise comparisons for annual biomass and carbon accumulation of seedlings in plantation and natural regeneration sites. The significant differences on biomass and carbon accumulation were observed at ages of 5, 6 and 7 ($P > 0.01$). Figure 4 shows that important values of biomass accumulation are observed at 4–7 years, especially in the plantation as a product of a greater tree-ring width growth in comparison to the naturally regenerated site. The estimates of biomass and carbon accumulation by tree age both for plantation and natural regeneration are shown in Table 4. After 7 years of evaluation, the seedlings in the plantation contain twice the biomass and carbon allocation than those evaluated in the natural regeneration. In the plantation, a 7-year-old tree has on average 4.38 and 2.21 kg of biomass and carbon

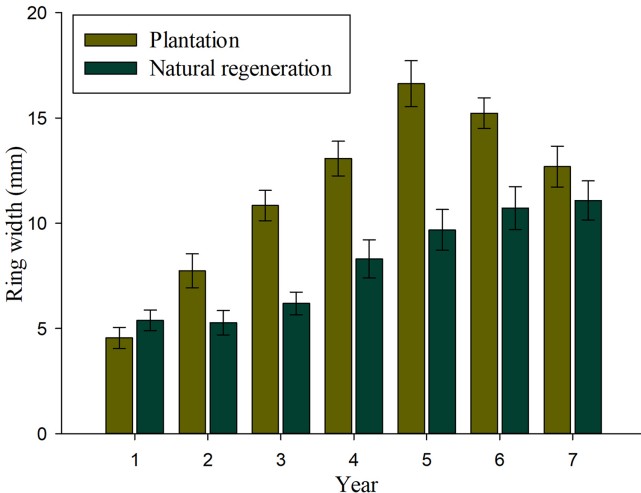

**Figure 3 Average values (bars) and standard error (whiskers) of the ring width per year recorded in *Pinus durangensis* seedlings from plantations and natural regeneration.**

**Table 3 Pairwise comparisons for the annual accumulation of biomass and carbon of seedlings in plantation and natural regeneration sites.**

| Comparison | Year | Biomass | | Carbon | |
|---|---|---|---|---|---|
| | | *t* ratio | *P* value | *t* ratio | *P* value |
| Plantation—Natural regeneration | 1 | −0.062 | 0.9506[ns] | −0.062 | 0.9506[ns] |
| Plantation—Natural regeneration | 2 | 0.057 | 0.9548[ns] | 0.057 | 0.9548[ns] |
| Plantation—Natural regeneration | 3 | 1.026 | 0.3064[ns] | 1.026 | 0.3064[ns] |
| Plantation—Natural regeneration | 4 | 1.766 | 0.0794[ns] | 1.766 | 0.0794[ns] |
| Plantation—Natural regeneration | 5 | 4.057 | 0.0001[***] | 4.057 | 0.0001[***] |
| Plantation—Natural regeneration | 6 | 5.307 | 0.0001[***] | 5.307 | 0.0001[***] |
| Plantation—Natural regeneration | 7 | 5.001 | 0.0001[***] | 5.001 | 0.0001[***] |

**Notes:**
[***] ($P < 0.001$).
ns, not significant.

content, respectively, whereas a naturally-regenerated seedling at the same age has on average 2.16 and 1.09 kg of biomass and carbon content, respectively (Table 4).

## CAI and MAI

The evolution of the CAI and the MAI with respect to the age of the seedlings in the plantation and the natural regeneration is shown in Fig. 5. Individual tree basal diameter increments both CAI and MAI were higher in the plantation at all analyzed ages in comparison to the naturally regenerated site. Trees from plantation exhibit an increasing linear trend on CAI at 1–5 years, and a decrease at 6 and 7 years (Fig. 5A), whereas seedlings from natural regeneration show a positive linear trend at all evaluated years (Fig. 5B). The maximum value of CAI in plantation seedlings was observed at 5 years, whereas this value in naturally regenerated seedlings is probably still not reached.

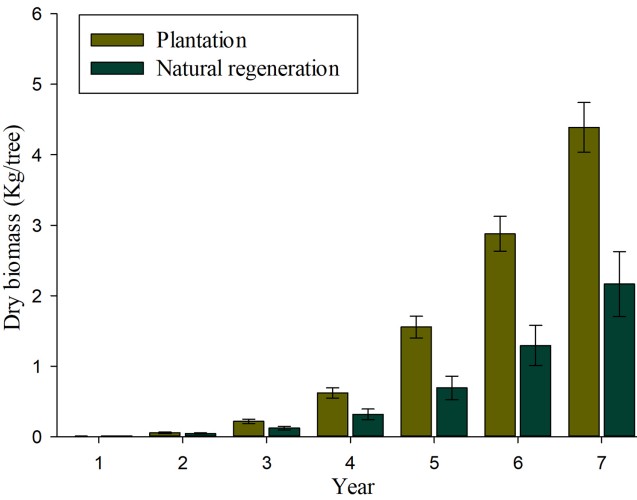

**Figure 4 Average values (bars) and standard error (whiskers) of the total aerial biomass accumulated in *Pinus durangensis* seedlings from plantation and natural regeneration.**

**Table 4 Annual average biomass and carbon accumulation in seedlings of *Pinus durangensis* in plantation and natural regeneration sites.**

| Year | Biomass (kg/tree) | | | | Carbon (kg/tree) | | | |
|------|------|------|------|------|------|------|------|------|
| | Plantation | | Natural regeneration | | Plantation | | Natural regeneration | |
| | Mean | SD | Mean | SD | Mean | SD | Mean | SD |
| 1 | 0.006 | 0.005 | 0.011 | 0.008 | 0.003 | 0.003 | 0.006 | 0.004 |
| 2 | 0.047 | 0.045 | 0.034 | 0.029 | 0.024 | 0.023 | 0.017 | 0.015 |
| 3 | 0.164 | 0.086 | 0.077 | 0.066 | 0.082 | 0.043 | 0.039 | 0.033 |
| 4 | 0.404 | 0.183 | 0.195 | 0.203 | 0.203 | 0.092 | 0.098 | 0.102 |
| 5 | 0.934 | 0.397 | 0.376 | 0.369 | 0.470 | 0.200 | 0.189 | 0.186 |
| 6 | 1.324 | 0.451 | 0.599 | 0.509 | 0.667 | 0.227 | 0.302 | 0.256 |
| 7 | 1.509 | 0.608 | 0.874 | 0.728 | 0.760 | 0.306 | 0.440 | 0.366 |
| Total | 4.388 | 1.776 | 2.165 | 1.912 | 2.210 | 0.894 | 1.090 | 0.963 |

**Note:**
SD, Standard deviation.

## DISCUSSION

In this article, we did not find significant effects in precipitation and temperature among the two studied sites, which make results on individual tree growth, biomass and carbon content comparable regarding these two environmental variables. However, even though precipitation and temperature are similar at both sites, it is well-known that these factors have a significant influence on the radial growth of tree species in Mexican forests (*Rodríguez Flores et al., 2014*; *Chávez-Gándara et al., 2017*).

The tree-ring width growth is the most used parameter to evaluate tree growing of forests (*Dobner, Huss & Tomazello Filho, 2018*). Both the kind and age of seedlings showed significant effects on annual tree-ring width growth. The planted seedlings showed a
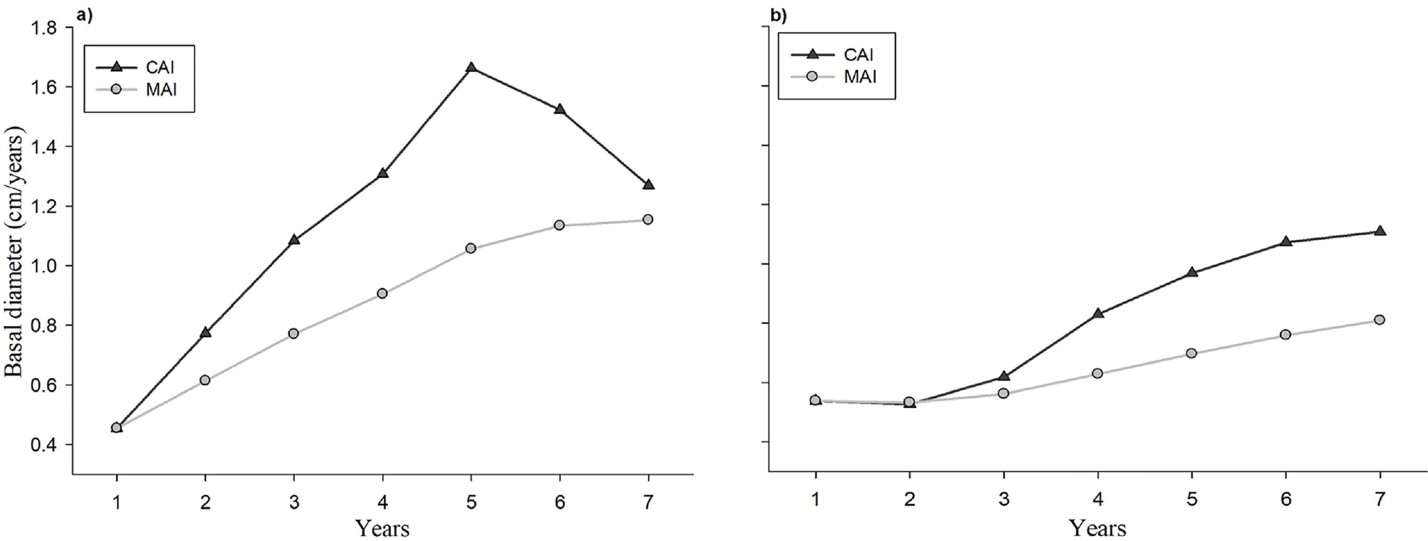

**Figure 5** Evolution of the current annual increment (CAI) and average annual increment (AAI) in basal diameter of the plantation (A) and natural regeneration (B).

constant increase from their establishment until age 5, while in the natural regeneration, this behaviour was maintained during all the evaluation periods. The decrease in tree-ring width growth observed in the plantation at 6–7 years may be due to the effect of competition that increases with stand age (*Soto-Cervantes et al., 2016*), especially in areas established with a high density like the one used in this work. On the other hand, the significantly lower tree density recorded in natural regeneration explains the constant increase on annual tree-ring width growth observed during the 7 years. Results on annual tree-ring width growth observed in the plantation of *Pinus durangensis* suggest that when a density of 2,500 plants ha$^{-1}$ is used, a pre-commercial thinning is needed at age 5 (1 year before tree growth reduction) to reduce tree competition and promote basal diameter growth. Studies have shown that a greater spacing between trees of pine species generates high availability of light and nutrients, which benefits growth both in diameter and height (*Baldwin et al., 2000*; *Arias, 2005*). The significantly higher tree-ring width growth recorded in the plantation at most evaluated ages may be also explained by the fact that in this site, seedlings have more light availability due to the absence of seed-trees in this site, and therefore, light is better assimilated to develop their maximum growth capacity and variations soil preparation activities during planting (*Plateros-Gastélum et al., 2018*; *Moretti et al., 2019*; *Landis, 2011*; *Pérez-Luna et al., 2020*). However, significant differences in width growth may also be partly due to variations in soil properties, slope, genetic differences, aspects that have not been evaluated in this study due to data limitations. According to *Måren et al. (2015)* tree growth is affected also by factors like climate, topography, aspect, inclination of slope and soil type. For example, tree growth is expected to be lower on steeply sloping sites than on flat or gently sloping sites because slope is one of the factors controlling the processes of infiltration of surface water into the subsurface (*Morbidelli et al., 2019*).

Biomass and carbon allocation in forests are important elements for timber production, carbon sequestration and the generation of other environmental services (*Graciano-Ávila et al., 2019*). In this article both the studied sites and seedlings age show significant differences in annual biomass and carbon accumulation ($P < 0.01$). The significantly higher annual biomass and carbon accumulation observed in the plantation trees may be due to the existence of a higher light environment which produces a high photosynthetic ability and favor growth in seedlings in comparison to the natural regeneration that has still seed-trees producing shadow (*Poorter et al., 2012*). However, the observed significant differences in biomass and carbon may be also caused by varying site conditions which greatly depend on soil quality, unfortunately such a soil data was not available for the present study. Results show that a 7-year-old seedling growing in the plantation produces an average of two times more biomass and carbon (*i.e.*, 4.38 and 2.21 kg, respectively) than a seedling of the same age growing in the natural regeneration (2.16 and 1.09 kg). Considering that the current stand density is of 1,881 trees ha$^{-1}$, the potential for carbon sequestration of the studied 7-year-old plantation of *Pinus durangensis* is of 4.15 tons ha$^{-1}$. These estimates of biomass and carbon content are lower than the estimations reported by *Pacheco-Escalona et al. (2007)* for a 6-year-old plantation of *Pinus greggii* Englem. With averages of 8.0 and 4.08 kg per tree, respectively. Higher biomass and carbon content in the *Pinus greggii* plantation may be a reflection of the use of higher number of seedlings per hectare during planting (4425), a better growth rate, and site conditions (*Rodríguez-Vásquez et al., 2021*).

The study of the current annual increment (CAI) and the mean annual increment (MAI) allow to define the optimum harvest age if the management objective is to maximize long-term yield (*Santiago-García et al., 2015*; *Cardalliaguet et al., 2019*). In this study, the maximum CAI point in basal diameter of seedlings in the evaluation period occurred at year 5 in the plantation, whereas in the natural regeneration it had not occurred yet. Moreover, it was observed that the curves of the CAI and MAI in basal diameter increment had not meet during the evaluation period, which indicate that is not yet possible to know the stand's optimal harvest age (age at which CAI = MAI), and therefore, new studies at older ages both in plantation and natural regeneration for the studied tree species are required. This result agrees with the study of *Mejía-Bojórquez, García Rodríguez & Muñoz Flores (2015)* who mention that for forest plantations of *Pinus durangensis*, younger than (15-years old), the intersection between the CAI and MAI curves has been not reported.

## CONCLUSIONS

The plantation and natural regeneration seedlings studied in this work showed significant differences on annual tree-ring width growth and annual biomass and carbon accumulation, the plantation exhibiting higher rates at most evaluated ages. Better tree growth rates observed in plantation seedlings are attributed mainly to silvicultural practices but can be also caused by variations in soil properties, slope and genetic differences, aspects that have not been evaluated in this study due to data limitations. Both kind of seedlings still do not show their maximum growth because the lines of CAI and MAI in basal diameter still do not intersect. The study reveals that forest plantations of

*Pinus durangensis* can be successfully established in the forests of Durango, Mexico, favoring both $CO_2$ capture among other ecosystem services.

## ACKNOWLEDGEMENTS

The authors are grateful to Andrea Losoya Simental for help with English language editing.

### Funding

This work was supported by Consejo Nacional de Ciencia y Tecnología (CONACYT-No. 297160). The funders had no role in study design, data collection and analysis, decision to publish, or preparation of the manuscript.

### Grant Disclosures

The following grant information was disclosed by the authors:
Consejo Nacional de Ciencia y Tecnología (CONACYT): 297160.

### Competing Interests

The authors declare that they have no competing interests.

### Author Contributions

- Jesús Alejandro Soto-Cervantes conceived and designed the experiments, performed the experiments, analyzed the data, prepared figures and/or tables, authored or reviewed drafts of the article, and approved the final draft.
- José Javier Corral-Rivas conceived and designed the experiments, performed the experiments, analyzed the data, prepared figures and/or tables, authored or reviewed drafts of the article, and approved the final draft.
- Pedro Antonio Domínguez-Calleros conceived and designed the experiments, performed the experiments, prepared figures and/or tables, authored or reviewed drafts of the article, and approved the final draft.
- Pablito Marcelo López-Serrano performed the experiments, prepared figures and/or tables, authored or reviewed drafts of the article, and approved the final draft.
- Eusebio Montiel-Antuna analyzed the data, prepared figures and/or tables, authored or reviewed drafts of the article, and approved the final draft.
- Emily García-Montiel analyzed the data, prepared figures and/or tables, authored or reviewed drafts of the article, and approved the final draft.
- Alberto Pérez-Luna conceived and designed the experiments, performed the experiments, analyzed the data, prepared figures and/or tables, authored or reviewed drafts of the article, and approved the final draft.

### Data Availability

The raw data are available in the Supplemental File.

## Supplemental Information

Supplemental information for this article can be found online at http://dx.doi.org/10.7717/
peerj.14774#supplemental-information.

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
