# Peer review of "Comparison of carbon content between plantation and natural regeneration seedlings in Durango, Mexico"

_PeerJ, doi:10.7717/peerj.14774_

## Round 0.1 · original submission · Major Revisions

I agree with the reviewers on the interest of the addressed topic. However, the article needs substantial improvement before being published.

I agree with reviewer 2 that the manuscript would benefit from reviewing by a fluent English-speaking person.

Two of the reviewers raise objections related to the statistical treatment. It is important that authors contemplate the recommendations of these reviewers.

Please take into account all the comments by the reviewers in the revised version of the manuscript. The suggestions of reviewer 2 related to experimental design and statistical analysis are particularly relevant. Also, consider my own comments in the included annotated manuscript.

·

Basic reporting

Well written and supported paper. Limited objectives and data available, but enough for supporting results.

Experimental design

Simple, but enough for the purposes defined

Validity of the findings

Congruent with data

Additional comments

None

Reviewer 2 ·

Basic reporting

The topic of the article is very interesting and fits within the scope of PeerJ. However, although it can be seen that there is significant work behind it there are two major drawbacks preventing its acceptance in the current version: the first one is the small size of the sample used, which renders the conclusions obtained difficult to extrapolate. Increasing the sample is not an easy thing to do a posteriori but, at least, the authors should clarify this drawback in the conclusions.
The second one is the statistical analysis of the data obtained. We are dealing with time series and analyzing the annual growth of basal diameter independently does not seem to be the best option. An alternative that should be evaluated is the use of mixed models in which the annual growth in diameter is analyzed against a fixed factor such as the group of seedlings, including the basal diameter as a covariate, adding a random factor per tree and taking into account the temporal correlation between data from the same individual. This methodology is very easy to implement in R software and would not involve excessive work for the authors, improving the robustness of the results.
Finally, although I am not a native English speaker, I believe that a review of the final document by a fluent English speaker could make the document easier to read.

Experimental design

Previously commented

Validity of the findings

Previously commented

Reviewer 3 ·

Basic reporting

The article is clear. It is suggested to add some background information in the introduction (lines 45-49; 131). Regarding the structure, the text is ordered and it is suggested to make adjustments in the order of presentation of the results (line 153-190). Table 1 should review the wording and use of thousands separators (e.g Mean N). The content of the document is related to the hypothesis and proposed objectives.

Experimental design

The experimental design is suitable for the characteristics of the study area. It is suggested to complement the definition of the biomass estimation method (line 127-135). The hypothesis is well stated, but some objectives are written as activities (line 72-79)

Validity of the findings

Is suggested to complement the discussion with quantitative results from other studies related to the species, and particularly about how forest management could contribute to maintaining and increasing the value of forest plantations for carbon sequestration

Annotated reviews are not available for download in order to protect the identity of reviewers who chose to remain anonymous.

Reviewer 4 ·

Basic reporting

In general, the manuscript is well written, though there are a few spots that could be improved. For example, line 72 the sentence is a bit convoluted. Similarly line 107 is a bit hard to read. In line 109, I would call the study units ‘plots’ instead of ‘sites.’ Another careful review of the entire manuscript is likely to find a few more spots that can be improved.

I would encourage the authors to consider using standard silviculture terminology for the regeneration harvest. It appears the harvest was a shelterwood cut but perhaps it was a clearcut with reserves. Identifying the silvicultural approach would help readers because “regeneration felling with parent trees” is not standard terminology (at least in the US). Basal area measurements pre and post harvest would also be useful for the reader in addition to the percentages presented in line 101 and onward.

Personally, I would include fewer details about the methods in the abstract and perhaps some information about the field sites.

Experimental design

It would be useful to know the approximate age of the parent trees in the natural regeneration stand.

Another crucial piece of information is the mean number of seedlings per acre. It seems competition may be a key influence so know the density of seedlings would be helpful.

Statistical analysis – It does not appear that the authors adjusted for multiple comparisons. While I recognize this will take some work and require changes through the manuscript, it is an important step to ensure reliability of the findings. I doubt it will change the significance of the differences.

Validity of the findings

The authors should be cautious in asserting that the decrease in growth in the plantation in the sixth year (line 194) is due only to competition. While competition is a likely driver there could be others such as precipitation – as discussed for slow growth in the first year. I also think it would be worth considering the impact of reduced precipitation on the natural regeneration in years 4 and 5 – could that help drive growth differences?

The paragraph starting line 212 would benefit from a mention of how similar P. durangensis is to the other species mentioned for readers not as familiar with the species.

Line 215 – Does the density of the species refer to a high density of trees, high density of wood itself, or something else?

It would be worthwhile to explore similarities and differences to the stands studied in Montes de Oca-Cano et al. (2008) since it appears a similar investigation.

The paragraph starting line 224 is good.

Additional comments

I recommend this paper for publication with minor revisions. It is a good study with direct relevance to the world of forestry and carbon storage. The biggest issue I see is the apparent lack of any adjustment for multiple statistical comparisons. This should be possible to correct with some work but without drastic changes to the manuscript. The other revisions I have suggested are small change that will improve the manuscript without too much work.

---

## Round 0.2 · Major Revisions

The manuscript addresses tree growth, biomass and carbon accumulation at two nearby forest sites, one with tree plantation and the other with natural regeneration. However, even though the two sites are close together and the climate is similar, the slope is considerably different. This is a crucial factor that must influence significantly soil characteristics, soil water regime and tree growth, but this factor has not been taken into account in the study. It seems likely that differences attributed to the type of management (plantation or natural regeneration) are due, at least in part, to the different slope.

Moreover, information on soil is lacking (the classification of both soils as lithic Leptosols does not seem plausible).

I have raised these questions in my review of the previous version. For example, my last comment was: “What is not clear is if differences are related with plantation or regeneration or with other possible differences (genetic characteristics of the seedlings, sivicultural treatments, soil,...)”. The authors’ answer has been always: “Thank you for the observation. We have rewritten this paragraph for clarity”.

For these reasons, despite the reviewers' recommendations, I think that the manuscript still needs major revisions, considering the slope and soil properties.
I have attached an annotated manuscript with detailed comments. Please take also into account the comments of reviewer 3. As for the Spanish in lines 126-127, I do not think it is strictly necessary to translate to English the name of the Institution; it is up to you.

Reviewer 2 ·

Basic reporting

no comment

Experimental design

no comment

Validity of the findings

no comment

Additional comments

The authors have taken into account all the suggestions made and I consider that the article deserves to be published.

Reviewer 3 ·

Basic reporting

The wording of the document is clear and the English has improved compared to the initial version. Although the bibliographical references are updated, they must be supplemented in some sections of the text. The structure of the document is neat. The results are consistent with the objectives of the study. Requires discussion settings (comments in attached document)

Experimental design

The experimental design, generation and analysis of information is consistent with the objective of the study. The hypothesis is clear, and the results allow its verification. Statistical analysis is confusing. In addition, it is necessary to adjust processes that determine the result (for example, allometric equation, and the comment is in the attached document)

Validity of the findings

The findings are consistent with the objective and hypothesis of the study. However, the discussion must be supplemented (comments in the attached document).

Annotated reviews are not available for download in order to protect the identity of reviewers who chose to remain anonymous.

Reviewer 4 ·

Basic reporting

The authors have done a good job with the revisions and I think the manuscript is clearer.
(though I did notice what appears to be a typo in the second sentence of the conclusion: "Both king of seedlings still do..."

Experimental design

I appreciate the authors work to improve the statistical analysis.

Validity of the findings

I think the findings are valid and have been improved through the review process.

Additional comments

good work!
sorry my review was slow

---

## Round 0.3 · accepted · Accept

My main concerns have been addressed by the authors. Therefore, I consider the article acceptable for publication. However, I still think that the manuscript would benefit from review by an English editor.

SECTION EDITOR EDITS
LINE NO: / BEFORE / AFTER / [COMMENTS]

LINE 36: / diameter both / diameter for both / [.]
LINE 110: / 50-year-old / fifty-year-old / [.]